# Socioeconomic Status and Interest in Genetic Testing in a US-Based Sample

**DOI:** 10.3390/healthcare10050880

**Published:** 2022-05-10

**Authors:** EJ Dusic, Deborah J. Bowen, Robin Bennett, Kevin C. Cain, Tesla Theoryn, Mariebeth Velasquez, Elizabeth Swisher, Jeannine M. Brant, Brian Shirts, Catharine Wang

**Affiliations:** 1Institute of Public Health Genetics, University of Washington, Seattle, WA 98105, USA; theoryn@uw.edu; 2Department of Bioethics, University of Washington, Seattle, WA 98105, USA; dbowen@uw.edu; 3Genetic Counseling Graduate Program, University of Washington, Seattle, WA 98105, USA; robinb@uw.edu; 4Department of Biostatistics, University of Washington, Seattle, WA 98105, USA; cain@uw.edu; 5University of Washington Medical Center, University of Washington Seattle, WA 98195, USA; mbv3@uw.edu (M.V.); swishere@uw.edu (E.S.); shirtsb@uw.edu (B.S.); 6Clinical Science & Innovation, Billings Clinic, Billings, MT 59105, USA; jbrant@coh.org; 7Department of Community Health Sciences, Boston University School of Public Health, Boston, MA 02118, USA; clwang@bu.edu

**Keywords:** genetic testing, socioeconomic status, cancer

## Abstract

Cancer is a significant burden, particularly to individuals of low socioeconomic status (SES). Genetic testing can provide information about an individual’s risk of developing cancer and guide future screening and preventative services. However, there are significant financial barriers, particularly for individuals of low SES. This study used the Early Detection of Genetic Risk (EDGE) Study’s patient baseline survey (*n* = 2329) to evaluate the relationship between socioeconomic status and interest in pursuing hereditary cancer genetic testing. Analysis was completed for two interest outcomes—overall interest in genetic testing and interest in genetic testing if the test were free or low cost. Many demographic and SES variables were predictors for interest in genetic testing, including education, income, and MacArthur Subjective Social Scale (SSS). After controlling for the healthcare system, age, and gender, having a higher education level and a higher household income were associated with greater general interest. Lower SSS was associated with greater interest in genetic testing if the test was free or low cost. If genetic testing is the future of preventative medicine, more work needs to be performed to make this option accessible to low-SES groups and to ensure that those services are used by the most underserved populations.

## 1. Introduction

Cancer is a significant health burden to both individuals and the United States (U.S.) health care system. This disproportionately affects individuals in low-SES income groups [1,2,3,4], who have increased mortality and incidence of developing all types of cancers when compared to their high-SES counterparts [2,5]. This is true for cancers without strong genetic influences and cancers linked to inherited gene variants. Studies have found that individuals who report less than 12 years of education have a 42% increased risk of developing colorectal cancer compared to individuals who have received more than 12 years of formal education [6]. Similar discrepancies are seen for prostate and breast cancer mortality, for which individuals of low-SES have a 19% and 6% higher mortality rate, respectively [2]. This demonstrated disparity in cancer mortality between those of low- and high-SES has seemingly gotten worse over the past 60 years. Data from the 1950s reported that individuals of low-SES actually had lower cancer mortality rates compared to individuals of high-SES. Recent data from 2010–2014 showed that this is no longer true, with individuals of low-SES now having a 22% higher cancer mortality rate. Additionally, there remain SES disparities in survival rates for cancer patients, and these disparities exist even after controlling for cancer stage and tumor grade [2,7,8,9,10].

These high rates of cancer largely reflect social determinants of health, health behaviors that are influenced by those determinants, healthcare accessibility, and access to what Phelan and Link describe as ‘flexible resources’ [2,5,10,11]. Flexible resources include knowledge, social capital, and power prestige [11,12]. These resources shape individual health behaviors so that individuals of higher SES know about, have access to, and can afford more preventative and screening sources [12]. Subsequently there exist huge socioeconomic disparities in the way healthcare is delivered and in the quality of care that individuals receive. There exists a significant need for more access to preventative and screening measures for individuals of low-SES [2].

Genetic testing for germline hereditary cancers in a primary care setting is the future of preventative medicine [13]. Genetic testing can be a successful and effective tool for informing individual and familial risk as well as guiding uptake of preventative services, and is recommended at a federal level for understanding a patient’s risk of developing breast, ovarian, tubal, peritoneal, and colorectal cancer [14,15]. This is reflected in the addition of the topic area ‘Genomics’ in Healthy People 2020, which aims to increase the evidence supporting the use of genetic testing in guiding clinical decision-making and public health interventions more broadly [14]. 

Having information about an individual’s risk of developing cancer can guide future screening and preventative services, improve the quality of care a patient will receive, and reduce the incidence of cancer [13]. Additionally, once an individual knows their genetic test results, their relatives (who potentially have inherited the same pathogenic variant) may be able to pursue genetic testing at a reduced price or through insurance coverage. Still, preventative cancer measures are largely underutilized in historically marginalized populations [16]. Previous work has indicated that individuals of low-SES are less likely to utilize genetic testing and other preventative measures [17]. For example, individuals of low-SES are 17% less likely to be treated with biological and precision therapies for cancer [16]. 

This study aims to evaluate a possible mechanism for why cancer genetic testing is underutilized by individuals of low-SES. To do so, we examine the relationship between SES and interest in pursuing genetic testing using data from the Early Detection of GEnetic Risk (EDGE) [18]. Previous work in low-SES study populations has found individuals who make a low income are interested and motivated to participate in screening and prevention activities [19,20] and that income does not predict interest in genetic testing [21]. Other work has indicated that individuals of high-SES are more willing to undergo genetic testing for cancer risk [22]. However, a large proportion of this work was conducted more than 20 years ago, and several changes in genetic technology and accessibility of genetic testing have taken place in that time. Additionally, income itself does not fully capture socioeconomic status. It is critical to extend this work and continue to investigate the reasons why preventative services are underutilized in the most underserved populations.

Given perceived and real financial barriers low-SES individuals face when accessing genetic testing, we hypothesize that individuals of low-SES may be less interested in pursuing genetic testing generally than individuals of high-SES. We also hypothesize that when the test is offered for free or at low cost, there will be no differences in interest between individuals of low-SES and individuals of high-SES’s interest in genetic testing. The overarching study aims are as follows:Describe the demographic characteristics of the EDGE Study population;Test whether individuals of low-SES are more or less interested in pursuing genetic testing in general than individuals of high-SES;Test whether individuals of low-SES are more or less interested in pursuing genetic testing than individuals of high-SES when the test is free or low cost.

## 2. Materials and Methods

### 2.1. The EDGE Study

This study uses data from the Early Detection of Genetic Risk (EDGE) Study. The larger goal of the EDGE Study is to reduce the overall burden of hereditary cancers by implementing cancer risk genomic testing for high-risk individuals in the primary care setting [18]. EDGE is working with two healthcare networks; MultiCare, a healthcare system located in suburban Washington State, and Billings Clinic, a system primarily located in rural Montana and Wyoming. We recruited from six clinics from each of the healthcare settings. 

### 2.2. Patient Baseline Survey

The patient baseline survey was distributed to patients in participating clinics starting in January 2021, before the implementation of the EDGE program. SES measures such as total household income, education level, and health insurance were collected in this survey. However, SES is meant to capture an array of resources that an individual has available to them; not just limited to money but also including resources such as access to knowledge, power, and social connections [12]. For this reason, we also included the MacArthur Scale of Subjective Social Status in the survey developed by Nancy Alder and colleagues [23] (Appendix A). The goal of this scale is to measure an individual’s perception of their relative position in society.

Other SES and demographic variables measured in the patient baseline survey were age, gender, race, ethnicity, and household size. Age was a free response question, although patients were required to be over the age of 25 to participate. For gender, participants were given the options ‘Male’, ‘Female’, ‘Other’, or ‘Prefer not to answer’. If they selected ‘Other’, they were given a free-response option to specify their response. Participants could select as many race options as were applicable, including White, Black or African American, Asian, Native Hawaiian or Pacific Islander, Native American/American Indian/Alaskan Native, or Multiracial. Like gender, participants were also given an ‘Other’ option which if they selected, they could give a free response to specify. To measure ethnicity, participants were asked if they considered themselves Hispanic and/or Latino. 

For education, participants were given response options ranging from ‘Less than high school’ to ‘Graduate or professional degree’. Participants were given five response options for insurance, including ‘Commercial’, ‘Government/military insurance’, ‘Medicare’, ‘Medicaid’, or ‘No insurance’. They could select multiple responses here if applicable. For household size, participants could select one option ranging from one to ten or more. Lastly, for income, participants were given the option to select one of seven responses ranging from ‘Less than $15,000′ to ‘More than $150,000′. They were also given the option of ‘Prefer not to answer’. To measure interest, we developed a module (Appendix A) based on a measure published by Desrosiers et al. [24] This module asked questions about what would influence an individual’s decision to pursue genetic testing and overall interest in genetic testing. Responses were formatted as a five-point Likert scale.

### 2.3. Recruitment

Patients from the two healthcare systems were asked to participate in the patient baseline survey. We reached out to a total of 6588 patients across all twelve clinics—549 from each clinic. Our original intent was to be able to recruit at least 200 from each clinic. A total of 2329 patients responded, an average of 194 per clinic. Patients who did not have an email or did not want to complete the survey electronically were mailed a copy of the survey to complete and mail back. Participants were compensated with a $10 Tango gift card for completing the survey. 

### 2.4. Data Analysis

Data from the patient baseline survey was analyzed in SPSS. Demographic variables were checked for outliers and data was cleaned when necessary. We removed three outliers in age that were likely due to mistakes in data entry. Missing values were not replaced. We used descriptive statistics to describe income, education level, health insurance, the MacArthur scale, and race and ethnicity for our population of interest. 

Analyses were run using two outcomes: overall interest in pursuing genetic testing and interest in pursuing genetic testing if the test were free or low cost. Because both outcome variables had irregular distributions, we split them into binary variables where people who responded four or higher were ‘high interest’ and those who responded less than four were ‘low interest’. We also recoded all demographic and SES variables into binary variables. ‘Prefer not to answer’ responses were excluded from analysis. If a case did not include one of the variables included in analysis, the case was excluded. For gender, ‘Other’ gender responses (*n* = 3) were excluded from analysis because there were so few. For the same reason, race and ethnicity were combined into one variable, so that White, non-Hispanic participants represented one group (*n* = 2011) and non-White and/or Hispanic participants represented the second (*n* = 217). Race and ethnicity were also analyzed as individual variables such that each possible race and ethnicity response was its own independent variable. This did not change the results of the analysis.

Education groups were split at those who had received at least an associate degree or higher (*n* = 1431) and those who had received some post-high school training or less (*n* = 784). Insurance was split between those who had commercial insurance (*n* = 1378) and those who only had Government/military insurance, Medicare, Medicaid, or no insurance (*n* = 836). No insurance was grouped with other non-commercial insurance because there remain disparities in insurance coverage of cancer genetic testing for underserved populations [25]. Participants who had both commercial insurance and other forms of insurance were only counted as having commercial insurance. Individuals who reported a household size of two or less were grouped (*n* = 1515) and individuals reporting three or more were grouped (*n* = 571). For household income, individuals making less than $74,999 were grouped (*n* = 957) and individuals making at least $75,000 were grouped together (*n* = 1243). Lastly, MacArthur Scale of Subjective Social Status responses were split so that individuals scoring six or higher were grouped together (*n* = 1453) and those scoring five or below were grouped (*n* = 728). 

We used binary logistic regression models to assess the relationship of each of the SES and demographic variables and both outcomes. We ran three logistic regression models for every combination of outcome variable and predictor variable: the first one was the simple association between predictor and outcome controlling for nothing, the second controlled for healthcare system and the third model controlled for healthcare system, age, and gender.

## 3. Results

### 3.1. Study Population Characteristics

A total of 2329 people completed the patient baseline clinic, with 1312 coming from Billings Clinic and 1017 coming from MultiCare Health System. The median age of all participants was 61 years with a standard deviation of approximately 15 (Table 1). A total of 61% of participants identified as female and 37% as male. The majority (approximately 88%) of participants identified as White and Not Hispanic or Latino. 

Most participants were at least high school graduates, with only 3% of our study population falling below this education line. Twenty percent of participants had received a graduate or professional degree. Most participants had a household size between one and two individuals (*n* = 1585). Results show a relatively even distribution for household income, with the minority of participants making less than $25,000 a year (*n* = 294). While 61% of participants indicated that they had commercial/private insurance (*n* = 1411), nearly half of participants reported having Medicare or Medicare supplement (*n* = 1073). An additional 20% had either government/military insurance, Medicaid, or no insurance (*n* = 455). Lastly, for the MacArthur Scale of Subjective Social Status, most participants considered themselves between a five or seven (*n* = 1338). A smaller amount rated themselves as eight or higher (*n* = 562) and an even smaller amount rated themselves four or below (*n* = 359).

### 3.2. Demographic and SES Variables Are Significantly Associated with Interest

Odds ratios (OR) for most SES and demographic variables were similar for both clinic systems, with slight variation in education and household size. All three demographics variables—age, gender, and race and ethnicity—were significantly associated with general interest in genetic testing (Table 2). Unexpectedly, individuals who were 65 or older had a lower rate of reporting high interest in genetic testing (OR 0.55, 95% CI: 0.46–0.66). Men also had much lower odds of reporting high interest in genetic testing compared to women (0.60, 95% CI: 0.50–0.71). Interestingly, individuals who are not White and/or Hispanic had much higher odds of reporting high interest in genetic testing compared to White and non-Hispanic participants. White and non-Hispanic participants had a 0.62 lower odds of reporting high interest in genetic testing compared to individuals who identified as not White and/or Hispanic (95% CI: 0.46–0.83). Household size was also significantly associated with interest, individuals with a household size of 2 or more had 1.30 greater odds of reporting high interest in genetic testing (95% CI: 1.07–1.57).

Out of the remaining four SES variables, two were significantly associated with general interest in genetic testing. Individuals from both healthcare systems who received an associate’s degree or higher had greater odds of reporting interest in genetic testing (1.31, 95% CI: 1.11–1.56). Additionally, participants who reported a household income of less than $75,000 had 1.48 greater odds of indicating low general interest in genetic testing (95% CI: 1.24–1.78). While the remaining two variables, insurance and subjective social status, were not significantly associated with general interest, they were near significant where *p* = 0.13 and *p* = 0.14, respectively. 

### 3.3. More Interest in Genetic Testing If the Test Was Free or Low Cost

As one would expect, individuals were much more likely to report high interest in genetic testing if the test was free or low cost. For general interest in genetic testing, the percentage of individuals who reported high interest within in each demographic and SES variable ranged from 45.8% to 65% (Table 2). However, the percentage of people who reported high interest in genetic testing if the test was free or low cost within each variable ranged from 77.4% to 88.4% (Table 3). This difference was particularly true for income. For general interest, among individuals with household incomes less than $75,000 per year, 51.3% indicated high interest in genetic testing. This percentage changes to 83.3% if the test was offered for free or low cost. The same pattern was true for all demographic and SES variables. In total, among participants who answered both interest questions (*n* = 2277), 31.7% indicated low general interest in genetic testing in general and high interest in genetic testing if the cost were free or low cost.

Fewer demographic and SES variables were significant predictors of interest in genetic testing if the test were free or low cost (Table 3). The ORs for age and gender were nearly the same among the two interest outcomes. The one SES variable that was significant for the free or low-cost outcome was the MacArthur Scale of Subjective Social Status. Individuals who rated themselves as less than or equal to 5 on the subjective social scale 0.52 times lesser odds of reporting low interest in genetic testing (95% CI: 0.41–0.68). 

### 3.4. Interest after Controlling for Healthcare System, Age, and Gender

Most associations between general interest in genetic testing and SES/demographic variables remained the same after controlling for healthcare system (Table 4). After controlling for healthcare system, age, and gender, only education and household income had significant associations with general interest. Odds ratios for these variables were nearly identical to ORs calculated in Table 2. Like the results in Table 2, the MacArthur Scale of Subjective Social Status was the only significant SES indicator of interest in genetic testing dependent on if the test is free or low cost after controlling for system, age, and gender (Table 5). Household size also had a strong relationship with interest in free or low-cost genetic testing when only controlling for healthcare system (OR = 1.30, 95% CI: 1.00–1.69).

## 4. Discussion

This project aimed to examine the relationship between SES and interest in pursuing genetic testing for hereditary cancers. This was accomplished through collecting and analyzing data in our patient baseline survey. We hypothesized that individuals of low-SES may be less interested in pursuing genetic testing than individuals of high-SES. My analysis showed that for most variables, there were differences in general interest in genetic testing. However, for some indicators individuals of low SES were more likely to be interested in genetic testing and for other indicators individuals of high SES were more likely to be interested in genetic testing. For example, individuals who rated themselves lower on the MacArthur Scale of Subjective Social Status were more interested in pursuing genetic testing but individuals who reported a household income lower than $75,000 were less interested in genetic testing compared to individuals from high-income households. 

Individuals of high SES were more interested in genetic testing in general when compared to individuals of low SES. Given the results of this analysis, it may be possible that individuals who make higher income or have higher education levels may be more interested in cancer genetic testing because they perceive fewer barriers to genetic testing. Overall, all individuals were much more interested in genetic testing if the test was free or low cost. This finding aligns with previous work that indicates that individuals perceive cost as a barrier to genetic testing. More importantly, it highlights the need for implementation of accessible genetic testing interventions.

While genetic testing can be a successful tool for preventing cancer incidence, there remain significant financial barriers. Healthcare inequalities between the wealthiest and poorest individuals in the United States have continued to rise over the last seventy years [2]. The National Cancer Institute’s website cites the cost of genetic testing itself, if not covered by insurance, as a potential harm to the individual undergoing genetic testing [26]. The out-of-pocket cost of genetic testing can be up to several thousand dollars, depending on the type of test and if their insurance will cover the cost of genetic testing [27].

There is limited research on understanding the financial barriers of genetic counseling and genetic testing [28]. However, evidence suggests that both patients and non-genetic providers see cost as a barrier to pursuing genetic testing. Our own work from the EDGE Study (manuscript in progress) is finding literacy, cost, and SES influence clinician behavior in whether or not they will refer a patient to genetic services. Costs associated with the process of accessing genetic services can be an additional barrier to care. For example, one study found that women with low incomes may be less likely to pursue genetic testing for hereditary breast and ovarian cancer because they may have a less flexible work hours and less reliable mode of transportation [28]. Lastly, even after an individual receives a positive genetic variant test result, there are barriers to access of screening and preventative services by SES [2,29]. There is a great need for the implementation of both accessible genetic testing services and accessible cancer prevention services, particularly for vulnerable populations. 

One major limitation to this project was the racial and ethnic diversity of the study population. Eighty-nine percent of our study population identified as white and 88% identified as non-Hispanic or Latino. According to the 2020 U.S. Census Bureau data for the distinct regions where these clinics are located, 58.9–94.2% of the population identify as white and 50.5–92.2% as non-Hispanic or Latino [30]. Although our findings suggest that individuals who are not White and/or Hispanic were more interested in genetic testing, further analysis with a more inclusive study population would be necessary to draw any conclusions. As with every other health outcome that disproportionately impacts individuals of low-SES and non-White individuals, it is important to consider the contribution of race and how it may impact interest in pursuing genetic testing. An emerging body of work has documented how physician’s racial biases impact referrals to genetic testing services [31]. Further work with a different study population would need to be performed to investigate racial disparities and uptake in genetic testing services.

Other limitations include a low response rate to our survey (approximately 34%, which limits the generalizability of these findings) and the geographical focus. As previously mentioned, we attempted to control for geographical differences in our analysis but only used clinics from two geographically distinct areas. Additionally, although this survey used MacArthur Scale of Subjective Status to help capture a full picture of SES, it showed opposite results compared to other SES variables. SES remains a difficult concept to measure; although indicators such as income, education, and subjective social status are clearly related, no single variable paints a full picture of SES. Of note, 6.7% of individuals who completed the survey reported already having had genetic testing for hereditary cancer while 3.7% were uncertain if they have had genetic testing for hereditary cancer and 88.6% reported not having had genetic testing. Additionally, 22.6% of individuals who had taken the survey had already been diagnosed with cancer at some point in their life. Research has found that individuals who have had cancer are interested in pursuing genetic testing to find out more about their family members’ risk of developing cancer or finding out if they should procure additional preventative screenings [32]. It is possible that this was influential of our findings.

Lastly, understanding interest in genetic testing is significant because it could ultimately influence whether an individual will decide to pursue genetic testing. In Ajzen’s theory of planned behavior [33], an attitude towards a behavior is a large contributor to the formation of a behavioral intention. While there is limited research that demonstrates this relationship with genetic testing, there is some evidence to support the relationship between interest and uptake in genetic testing. For example, in a study that looked at genetic testing interest and uptake in genetic testing for patients with lung cancer, it was found that participants who said they “definitely would” take a genetic test were more likely to take a genetic test [34]. However, this effect was modest and more research supporting this relationship needs to be conducted to back this claim.

## 5. Conclusions

If genetic testing for hereditary cancers is to become the future of preventative medicine, it is necessary that it first becomes more accessible to individuals of low-SES. This is particularly paramount given that individuals of low-SES have higher cancer incidence and mortality. Future work investigating this relationship should look at SES and uptake in genetic testing when it is offered as a free service. Additional work could investigate the relationship between interest in pursuing genetic testing and uptake. This is a largely understudied area, and although there are theoretical models that support this relationship, research has shown mixed results. Altogether, major financial barriers exist that perpetuate poor health outcomes between the poorest and the richest individuals in the U.S. More work needs to be performed to provide preventative health service to low-SES groups and to ensure that those services are used by the most vulnerable, relevant groups. 

## Figures and Tables

**Table 1 healthcare-10-00880-t001:** Characteristics of the EDGE Patient Baseline Survey participant.

	Study Participants (*n* = 2329)	
Characteristics	No. of Participants	%
**Age, years**		
Median (SD)	61 (15.3)	-
**Gender**		
Male	858	37%
Female	1428	61%
Other	3	0%
**Race ***		
White	2079	89%
Black or African American	41	2%
Asian	67	3%
Native Hawaiian or another Pacific Islander	7	0%
Native American/American Indian/Alaskan Native	42	2%
Multiracial	36	2%
Other	24	1%
Prefer not to answer	58	2%
**Ethnicity**		
Not Hispanic or Latino	2041	88%
Hispanic or Latino	58	2%
Prefer not to answer	84	4%
**Education**		
Less than high school	18	1%
Some High School, no diploma	45	2%
High school graduate	275	12%
Some post high school training, no degree or certificate	499	21%
Associate college degree, or completed occupational, technical, or vocational program and received degree or certificate	342	15%
Bachelor’s degree	652	28%
Graduate or professional degree	463	20%
**Insurance ***		
Commercial (private) insurance	1411	61%
Government/military insurance	200	9%
Medicare	1073	46%
Medicaid	207	9%
None	48	2%
**Household Size ****		
One	466	20%
Two	1119	48%
Three	264	11%
Four or more	320	14%
**Household Income**		
Less than $15,000	123	5%
Between $15,000 and $24,999	171	7%
Between $25,000 and $49,999	335	14%
Between $50,000 and $74,999	382	16%
Between $75,000 and $99,999	339	15%
Between $100,000 and $149,999	329	14%
More than $150,000	259	11%
Prefer not to answer	345	15%
**MacArthur Scale of Subjective Social Status ****		
One through four	359	15%
Five	404	17%
Six	448	19%
Seven	486	21%
Eight through ten	562	24%

* Participants had the option to select more than one response. Participants who selected more than response were counted for each box they checked. ** Participants were given the selection one through ten on the survey, but responses have been grouped for analytic purposes.

**Table 2 healthcare-10-00880-t002:** Bivariate analysis for interest outcome 1.

*If Your Personal and Familial History Suggested You Were at High Risk for Cancer, How Interested Would You Be in Having Genetic Testing?*					
**Combined MultiCare and Billings**					
**Characteristics**	**Low Interest (<4) Count (%)**	**High Interest (≥4** **) Count (%)**	**Chi-Square**	** *p* ** **-Value**	**OR (95% CIs)**
**Age**			44.5	<0.001 **	0.55 (0.46–0.65)
<65 years old	422 (39.6%)	645 (60.4%)			
≥65 years old	537 (54.2%)	453 (45.8%)			
**Gender**			35.4	<0.001 **	0.60 (0.50–0.71)
Female	583 (41%)	838 (59%)			
Male	458 (53.9%)	392 (46.1%)			
**Race and Ethnicity**			10.6	0.001 **	0.62 (0.46–0.83)
Non-White and/or Hispanic	76 (35%)	141 (65%)			
White and Non-Hispanic	930 (45%)	1066 (55%)			
**Education**			9.7	0.002 **	1.31 (1.11–1.56)
Lower than Associate’s Degree	417 (50.1%)	416 (49.9%)			
Associate’s Degree or Higher	628 (43.3%)	822 (56.7%)			
**Insurance**			2.4	0.125	1.14 (0.96–1.35)
Only No Insurance or Non-Commercial Insurance	430 (47.9%)	468 (52.1%)			
Commercial Insurance	626 (44.6%)	777 (55.4%)			
**Household Size**			7.1	0.008 **	1.30 (1.07–1.57)
≤2	746 (47.4%)	828 (52.6%)			
>2	238 (41.0%)	343 (59.0%)			
**Household Income**			18.2	<0.001 **	1.48 (1.24–1.78)
<$75,000	487 (48.7%)	514 (51.3%)			
≥$75,000	361 (39.0%)	565 (61.0%)			
**MacArthur Scale of** **Subjective Social Status**			2.2	0.136	0.88 (0.73–1.04)
≤5	330 (43.4%)	431 (56.6%)			
>5	694 (46.7%)	793 (53.3%)			

** Statistically significant, where *p* < 0.01.

**Table 3 healthcare-10-00880-t003:** Bivariate analysis for interest outcome 2.

*I Would Be Interested in Genetic Testing If the Test Was Free or Low Cost*					
**Combined MultiCare and Billings**					
**Characteristics**	**Low Interest (<4) Count (%)**	**High Interest (≥4)** **Count (%)**	**Chi-Square**	** *p* ** **-Value**	**OR (95% CIs)**
**Age**			30.2	<0.001 **	0.53 (0.42–0.66)
<65 years old	141 (13.3%)	923 (86.7%)			
≥65 years old	220 (22.5%)	756 (77.5%)			
**Gender**			23.8	<0.001 **	0.58 (0.47–0.72)
Female	204 (14.5%)	1203 (85.5%)			
Male	191 (22.6%)	655 (77.4%)			
**Race and Ethnicity**			0.1	0.73	0.94 (0.64–1.37)
Non-White and/or Hispanic	35 (16.2%)	181 (83.8%)			
White and non-Hispanic	339 (17.1%)	1640 (82.9%)			
**Education**			0.4	0.509	0.96 (0.74–1.16)
Lower than Associate’s Degree	138 (16.8%)	681 (83.2%)			
Associate’s Degree or Higher	259 (17.9%)	1184 (82.1%)			
**Insurance**			1.2	0.281	1.13 (0.91–1.41)
Only No Insurance or Non-commercial Insurance	166 (18.7%)	722 (81.3%)			
Commercial Insurance	236 (16.9%)	1158 (83.1%)			
**Household Size**			3.7	0.055	1.29 (0.99–1.68)
≤2	287 (18.4%)	1273 (81.6%)			
>2	86 (14.9%)	493 (85.1%)			
**Household Income**			0.002	0.964	1.00 (0.78–1.26)
<$75,000	166 (16.7%)	827 (83.3%)			
≥$75,000	155 (16.8%)	768 (83.2%)			
**MacArthur Scale of** **Subjective Social Status**			25.2	<0.001 **	0.52 (0.41–0.68)
≤5	88 (11.6%)	668 (88.4%)			
>5	297 (20.1%)	1179 (79.9%)			

** Statistically significant, where *p* < 0.01.

**Table 4 healthcare-10-00880-t004:** Multiple logistic regression of interest outcome 1 controlling for healthcare system, age, and gender.

*If Your Personal and Familial History Suggested You Were at High Risk for Cancer, How Interested Would You Be in Having Genetic Testing?*						
	** Controlling for Healthcare System **			** Controlling for Healthcare ** ** System, Age, and Gender **		
	**OR (95% CIs)**	**S.E.**	**Sig.**	**OR (95% CIs)**	**S.E.**	**Sig.**
Race and Ethnicity	0.67 (0.50–0.90)	0.15	0.008 **	0.75 (0.55–1.04)	0.17	0.085
Education	1.28 (1.08–1.52)	0.09	0.005 **	1.30 (1.08–1.57)	0.09	0.005 **
Insurance	1.14 (0.97–1.35)	0.09	0.12	1.06 (0.88–1.28)	0.1	0.54
Household Size	1.29 (1.06–1.56)	0.10	0.01 **	1.03 (0.82–1.28)	0.12	0.83
Household Income	1.44 (1.20–1.73)	0.09	<0.001 **	1.60 (1.31–1.96)	0.09	0.002 **
MacArthur Scale of Subjective Social Status	0.87 (0.73–1.04)	0.09	0.13	1.07 (0.89–1.30)	0.1	0.47

** Statistically Significant, where *p* < 0.01. Each indicator has the same base category as in Table 2 and Table 3.

**Table 5 healthcare-10-00880-t005:** Multiple logistic regression of interest outcome 2 controlling for healthcare system, age, and gender.

*I Would Be Interested in Genetic Testing If the Test Was Free or Low Cost*						
	** Controlling for Healthcare System **			** Controlling for Healthcare ** ** System, Age, and Gender **		
	**OR (95% CIs)**	**S.E.**	**Sig.**	**OR (95% CIs)**	**S.E.**	**Sig.**
Race and Ethnicity	0.90 (0.61–1.32)	0.20	0.58	1.21 (0.81–1.82)	0.21	0.36
Education	0.94 (0.75–1.18)	0.12	0.60	0.97 (0.76–1.24)	0.13	0.83
Insurance	1.13 (0.91–1.40)	0.11	0.28	1.03 (0.81–1.31)	0.12	0.83
Household Size	1.30 (1.00–1.69)	0.13	0.05 *	0.98 (0.72–1.33)	0.16	0.88
Household Income	1.02 (0.80–1.29)	0.12	0.896	1.08 (0.83–1.40)	0.14	0.58
MacArthur Scale of Subjective Social Status	0.52 (0.41–0.68)	0.13	<0.001 **	0.62 (0.48–0.822)	0.14	<0.001 **

* Statistically significant, where *p* < 0.05. ** Statistically significant, where *p* < 0.01. Each indicator has the same base category as in Table 2 and Table 3.

## Data Availability

The data presented in this study are available on request from the corresponding author. The data are not publicly available at this time due to privacy restrictions. The EDGE study data will be made accessible via the NIH’s database after the project end date (August 2024).

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
