# Peer review of "Socioeconomic Status and Interest in Genetic Testing in a US-Based Sample"

_healthcare, 2022, doi:10.3390/healthcare10050880_

Round 1

Reviewer 1 Report

Minor revision is requested. The topic of the article is of great interest and the results are clear and match with preconceptions that we had in this field. However, there are two items that, in my opinion, can be improved:

  • In general: I consider important to mention the proportion of participants that have suffered from cancer or other genetic diseases, since they can be more aware of the relevance of being tested for illness predisposition.
  • I found very interesting the last two points of the survey; however, I do not find them developed in the manuscript. It would be interesting to know what is the trend between the participant in both senses.

Reviewer 2 Report

In abstract

1- “if the test were free/low cost” Grammar revision

In introduction

2- Try to avoid redundancy.

3- U.S abbreviation should be preceded first time mentioning by (United States

4- Language editing is recommended (eg. “individuals of  income are interested and motivated”  )

In results

5- Table 1. I wonder if you combined race and ethnicity in one item

In discussion

6- “Other limitations include a low response rate to our survey” why do you count low response rate one of limitations? Is it better to consider the number of responding participants?

Reviewer 3 Report

Dear authors,

First of all, I would like to congrats you on the outstanding piece of work that you have developed. It was a pleasure to read your paper and to understand deeply what you have done and your major contribution to science.

As a sociologist, I am glad to see that finally, science is considering that health status is strongly influenced by the social and economic environment and that those ones should be considered in future health public policies. It is a justice issue.

Concerning your paper, the introduction section is coherently presented and as well as the aims. The literature is very well synthesized and gives a nice overview of the topic discussed. It is very focused.

The material and methods section should be considered an example for future works. The steps presented and easy to follow and understand the rationale of the method. This should be the basis of each study for the readers being able to understand what the authors have done. The transparency of the methods presented is a good achievement in this study.

The results and discussion sections are the most interesting parts. The results are presented in a very easy way to read and match with the study aims. The authors have greatly answered their research questions and objectives and were able to demonstrate that SES is a strong predictor of cancer genetic tests. In a world facing the worsening of social inequalities, this is a finding of great importance that should be considered.

For the reasons above mentioned, I considered that this study presents a strong and important contribution to the field.

Congratulations!

Reviewer 4 Report

The topics addressed by the authors are very important and include the influence of socioeconomic status on willingness to undergo genetic testing for cancer.  The study used the Early Detection of GEnetic Risk (EDGE) Study’s patient baseline survey to evaluate the relationship between socioeconomic status and interest in pursuing hereditary cancer genetic testing. What is important, the n is 2329, therefore the statistical significance of this study is very high. The statistical analysis is performed well and the general conclusions are interesting for the readers of Healthcare.

I think the manuscript can be accepted for publication in the present form. I encourage Authors to read and check for typos etc in the manuscript and verify the passage of text below the bibliography (shouldn't it be removed/moved?)
